# Optical single-shot readout of spin qubits in silicon

Andreas Gritsch[1,2], Alexander Ulanowski[1,2], Jakob Pforr [1,2] & Andreas Reiserer [1,2] ✉

Small registers of spin qubits in silicon can exhibit hour-long coherence times and exceeded error-correction thresholds. However, their connection to larger quantum processors is an outstanding challenge. To this end, spin qubits with optical interfaces offer key advantages: they can minimize the heat load and give access to modular quantum computing architectures that eliminate cross-talk and offer a large connectivity. Here, we implement such an efficient spin-photon interface based on erbium dopants in a nanophotonic resonator. We demonstrate optical single-shot readout of a spin in silicon whose coherence exceeds the Purcell-enhanced optical lifetime, paving the way for entangling remote spins via photon interference. As erbium dopants can emit coherent photons in the minimal-loss band of optical fibers, and tens of such qubits can be spectrally multiplexed in each resonator, the demonstrated hardware platform offers unique promise for distributed quantum information processing based on scalable, integrated silicon devices.

In the last decades, quantum information processing has seen major advances, and first systems have outperformed classical devices in certain tasks[1,2]. Advanced solid-state systems in this context are superconducting circuits[1] and spin qubits in silicon[3–6]. The latter can operate at higher temperatures $>1\,\mathrm{K}$[7], enable very dense integration[8], and give access to long-term quantum memories[9]. So far, however, upscaling of such systems is hampered by cross-talk and heating caused by the required low-frequency electromagnetic control fields.

In contrast, using optical fields for qubit readout and control offers several key advantages: First, optical frequencies offer superior bandwidth, allowing for frequency-multiplexed addressing of hundreds of qubits in a few cubic micrometers[10]. Second, optical modes are not thermally populated even at ambient temperature, which enables long-distance transmission of quantum states over glass fibers and avoids the need for cryogenic attenuators whose heat load impedes up-scaling[11]. Third, optical fields can be confined to the nanoscale[12], eliminating cross-talk even between closely-spaced photonic components. Finally, fast and efficient detectors for optical fields are readily available and can be integrated on silicon chips[13] to enable rapid qubit measurements with high fidelity.

Consequently, the last years have seen intense international efforts towards combining spin qubits in silicon with optical initialization, control and readout via spin-to-optical conversion[14]. A first milestone in this direction was the optical observation of single spins of erbium dopants[15] or color centers[16]. Recently, integrating these emitters into nanophotonic resonators[17–19] has improved the efficiency of photon outcoupling. Here, we build on these advances to demonstrate the optical initialization and readout of a single erbium spin qubit.

## Results

### Erbium dopants in silicon

Compared to all other solid-state emitters[20], including color centers in silicon[16,18,19,21–23] and other materials[24–27], as well as quantum dots[28] and layered materials[29], erbium has a unique combination of advantageous properties: it is the only known emitter with long-lived and coherent spin ground states[30] that simultaneously offers lifetime-limited optical coherence[31] in the telecommunications C-band, where loss in glass fibers is minimal. Furthermore, the protection of its electronic states in the inner 4f-shell can lead to exceptionally narrow spectral diffusion

[1]TUM School of Natural Sciences, Department of Physics and Munich Center for Quantum Science and Technology (MCQST), Technical University of Munich, James-Franck-Str. 1, Garching, Germany. [2]Max-Planck-Institute of Quantum Optics, Hans-Kopfermann-Str. 1, Garching, Germany. ✉e-mail: andreas.reiserer@tum.de

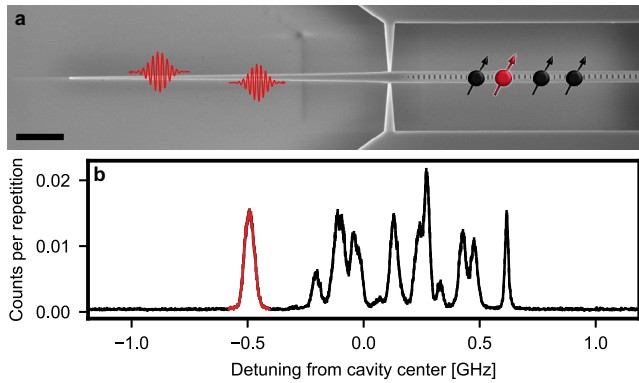

**Fig. 1 | Single erbium dopants coupled to a nanophotonic resonator. a** Several emitters (black and red spin symbols) are embedded at random positions within a nanophotonic resonator (SEM image, scale bar 3 μm). Photons (red curly arrows) are coupled in and out via a tapered feed waveguide that is brought in contact with a tapered glass fiber (not shown). **b** When the cavity frequency is tuned on resonance with erbium ensembles in site A[34] and the excitation laser frequency is varied, the fluorescence in the first 20 μs after 0.15 μs-long laser excitation pulses exhibits several sharp peaks within the cavity linewidth of 2.37(6) GHz FWHM (x-axis range). Each peak corresponds to a single erbium dopant, and several of them are well-separated from the others so that these dopants can be addressed individually. Error bars: 1 SD.

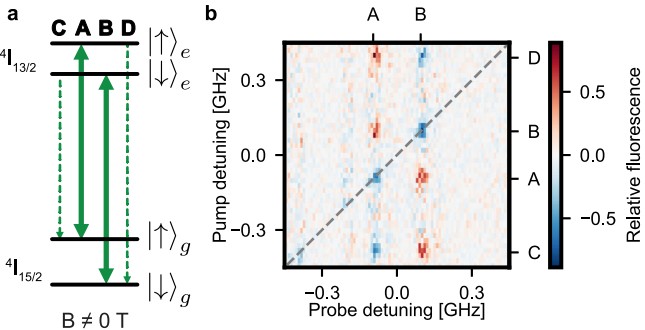

**Fig. 2 | Selective spin addressing. a** An external magnetic field lifts the degeneracy of the erbium spin levels, with a different splitting in the ground (g) and optically excited (e) states. Thus, the spin-preserving (bold arrows) and the spin-flip (dashed arrows) transitions can be excited selectively. **b** Relative fluorescence when the emitter is irradiated with pump and probe pulses of different frequency. Compared to the average signal without pump pulses, the fluorescence after resonant probe pulses is enhanced (red) or reduced (blue) by preceding pump pulses if the frequency of the latter is on resonance with one of the four optical transitions. The observed characteristic pattern allows for an unambiguous determination of the transition frequencies (A–D) and thus the effective g-factors in the ground and excited state. As a guide-to-the-eye, the dashed black diagonal line indicates identical pump and probe frequencies.

linewidths, down to 0.2 MHz[32,33]. This facilitates the spectral multiplexing of hundreds of emitters[10]. Finally, the slow spin-lattice relaxation rates in silicon[34] and other materials eliminate the need for sub-Kelvin temperatures, and thus for cryocoolers that use the rare isotope $^3$He. However, these advantages come at a price - compared to other emitters, in particular color centers and quantum dots, the optical transitions of erbium are much slower, with a typical timescale of several milliseconds in bulk crystals.

The resulting difficulty of low operation rates can be overcome by integrating the dopants into optical resonators[20]. The first experiments that in this way resolved single erbium emitters and demonstrated multiplexed spin control used either bulk resonators[32] or a hybrid approach[35], in which a silicon nanocavity is deposited on top of another host crystal. While both approaches are compatible with good spin- and optical coherence[32,33], they are incompatible with scalable nanofabrication. Therefore, in this work we instead integrate the dopants directly into silicon at the recently discovered site "A" that exhibits excellent optical properties at its emission wavelength of 1537.8 nm[34]: a short radiative lifetime in bulk (142 μs), and both narrow inhomogeneous (0.5 GHz) and homogeneous (<10 kHz) linewidths. These properties are preserved in commercial, wafer-scale nanofabrication processes[36], which offers unique prospects for up-scaling.

**Spin-photon interface**

To resolve individual erbium spins and facilitate their readout, we integrate them into a spin-photon interface based on a silicon photonic crystal resonator, as shown in Fig. 1a and further described in the Methods. Compared to our recent first observation of Purcell-enhanced spins in a similar device[17], we have improved the resonator design to reduce the mode volume almost twofold, to $0.83(\lambda/n)^3$, while maintaining a high quality factor of $82(3) \times 10^3$. At the same time, the novel cavity devices are close-to-critically coupled, such that we obtain a high photon outcoupling efficiency of ≈ 40% to the feed waveguide. In addition, an off-chip coupling efficiency of ≈ 50% is achieved using a tapered glass fiber[37]. The initial characterization of the device follows the techniques introduced in ref. 17 and described in Supplementary Note 1. The resulting spectrum is shown in Fig. 1b. For a more detailed investigation of the individual emitters, the cavity is

tuned precisely on resonance with each of them by varying the temperature (see "Methods"). By comparing their properties (see Supplementary Note 2), we find that the most red-detuned dopant (red) simultaneously exhibits the highest single-photon purity, $g^{(2)}(0) = 0.019(1)$, and the largest Purcell enhancement, $P = 177(2)$ in comparison to the bulk[34], which in turn results in the most favorable ratio between the lifetime-limited linewidth and the spectral diffusion linewidth among the dopants in this device. All measurements in this work are thus performed on this emitter. The obtained optical lifetime of 0.803(11) μs represents the fastest optical emission demonstrated for a single erbium dopant, almost ten-fold faster than that observed in previous experiments using other host crystals[33,38]. This will speed up all operations of the spin-photon interface, including optical spin readout and spin initialization.

To this end, an external magnetic field of 55 mT is applied along the (100) direction to split the effective spin-1/2 levels in the ground and optically excited states by different amounts[36], as shown in Fig. 2a. When this splitting exceeds the optical linewidth, a spin qubit encoded in the ground state levels can be initialized by cavity-enhanced optical pumping[20], i.e. by the irradiation of a laser that drives a spin-flip transition to a level whose spin-preserving decay is enhanced by the resonator. To determine the frequencies of the relevant transitions, we employ a pump-probe scheme with 500 consecutive pump pulses of 0.1 μs duration followed by a single 0.2 μs long probe pulse, after which we record the fluorescence signal $S(\nu_{pump}, \nu_{probe})$. For every probe laser frequency, we then average over all pump frequencies to determine the average signal $S_a(\nu_{probe}) = \langle S(\nu_{pump}, \nu_{probe}) \rangle_{\nu_{pump}}$. In Fig. 2b we then show the pump-induced fluorescence difference $S_d(\nu_{pump,probe}) = S(\nu_{pump}, \nu_{probe}) - S_a(\nu_{probe})$. As expected, a change is observed whenever the pump field is on resonance with any of the four transitions (A-D) and thus polarizes the spin. The obtained characteristic pattern allows therefore for unambiguous determination of the transition frequencies belonging to the same emitter. On the spin-flip transitions C and D, no significant signal is obtained after the probe pulses, in agreement with the expectation from the spin Hamiltonian[39]. This explains the absence of peaks in the right side of the figure. The faint features on the left side (around −0.2 and −0.4 GHz, close to transition C) are instead attributed to other emitters because they do not exhibit the characteristic pattern for all four pump frequencies.

## Spin properties

Next, we set the magnetic field to 0.3 T to further increase the separation between the levels. We then use spin polarization on either of the two spin-flip transitions followed by a fluorescence measurement on the spin-preserving line (A) to characterize the spin properties. We first insert a variable waiting time and measure a spin lifetime of 0.44(6) s, as shown in Fig. 3. We then implement optically detected magnetic resonance (ODMR) spectroscopy by driving the spin transitions using microwave (MW) pulses. The ODMR spectrum, Fig. 4a, exhibits a triple peak structure, which is well-fit by a sum of three Gaussian terms, each with a linewidth of 2.37(2) MHz FWHM. Such broadening is expected as the used crystal contains a natural abundance of 4.7% $^{29}$Si isotopes whose nuclear spins lead to a fluctuating magnetic field at the position of the erbium dopant. The observed line splitting is attributed to the coupling of the emitter to surrounding spins. At the used low dopant concentration, the estimated coupling to other erbium spins is too small to explain the observations. However, one expects that in our sample with its natural $^{29}$Si isotopic abundance, individual nuclear spins exhibit a superhyperfine coupling that is larger

than the broadening from the rest of the spin bath. A possible configuration is shown in Supplementary Fig. 2, albeit other configurations cannot be ruled out with certainty based on our measurements. In future experiments, such strongly-coupled nuclear spins may act as a quantum register, which is read and controlled via the erbium electronic spin − similar to earlier experiments with color centers in diamond[40] and other emitters in the solid state[27].

Next, we apply a resonant MW field for a varying duration. As shown in Fig. 4b, we obtain Rabi oscillations with $\pi$-pulse lengths of 2.3(2) $\mu$s and 1.6(2) $\mu$s when the MW is applied on the central peak or on the left side peak, respectively. Further measurements will be required to explain the increased oscillation frequency in the latter case. To further characterize the qubit coherence, we then apply a Hahn echo sequence on the central peak, in which a MW $\pi$-pulse is inserted between two $\pi/2$-pulses. As the available pulse power and bandwidth does not allow for a complete inversion, we subtract two measurements with opposite phases of the first $\pi/2$-pulse (Fig. 4c, inset). In this way, we obtain a Gaussian decay of the spin echo with $T_{\text{Hahn}} = 48(2)$ $\mu$s, as shown in Fig. 4c. To investigate the source of decoherence, we perform a second measurement at 4.45 K. Owing to condensed gas on the sample that shifts the cavity frequency, we can keep the optical transitions on resonance at this increased temperature. We find that the coherence time differs between the measurements at 2.75 and 4.45 K, which hints at paramagnetic impurities as a source of decoherence[27]. Still, our result is comparable to recent ensemble measurements of Er:Si with natural isotope abundance, and a strong improvement is thus expected in isotopically purified material at lower temperature[41]. Alternatively, dynamical decoupling may improve the qubit coherence time $T_2$ up to the lifetime limit of about one second once MW pulses with a higher Rabi frequency are implemented, e.g. using on-chip striplines[42].

## Optical single-shot readout

After characterizing the spin properties, we now turn to the optical single-shot readout via state-selective fluorescence[20]: A laser selectively excites one of the spin states, called the "bright" state, while the other state is detuned and thus stays "dark". Therefore, detecting a fluorescence photon heralds that the qubit is in the bright state. Because of the finite photon detection efficiency, 10(2)% in the current experiment, the dopant needs to emit several photons so that one is detected before the spin state is flipped. Thus, one requires a high

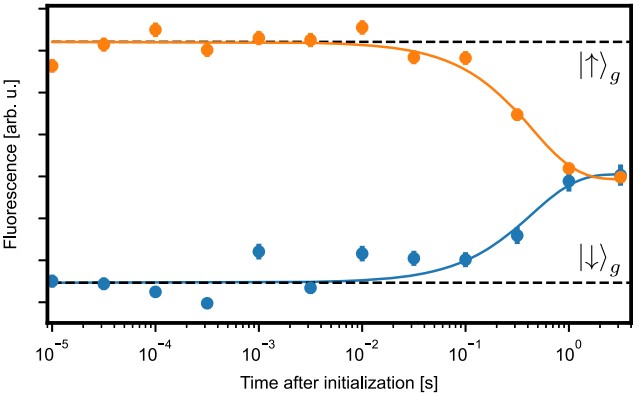

**Fig. 3 | Spin lifetime.** The spin is initialized by optical pumping on the spin-flip transitions and then probed after a variable waiting time by measuring the fluorescence integrated over 39 laser pulses. The spin lifetime is determined from an exponential fit of the fluorescence decay (averaged over 1000 repetitions for the bright state and 250 repetitions for the dark state). The statistical error bars (1 SD) do not include the finite initialization fidelity. During this measurement, the sample was kept at a temperature of 2.8 K and at a magnetic field of 0.3 T.

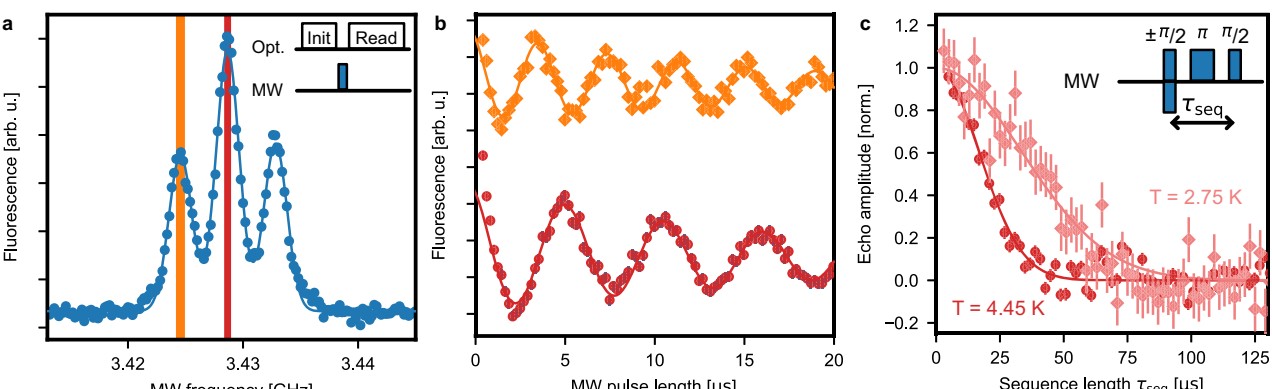

**Fig. 4 | Coherent control of the electronic spin qubit. a** For optically detected magnetic resonance spectroscopy, the spin is initialized before a microwave (MW) pulse is applied, followed by an optical readout sequence (see inset). The splitting into three lines, each with a FWHM of 2.37(2) MHz (solid Gaussian fit curve), is attributed to the coupling to proximal $^{29}$Si nuclear spins. **b** The spin population undergoes Rabi oscillations as a function of the MW pulse duration, both on resonance with the central peak (red) and one of the side peaks (orange).

The Rabi frequency extracted from an exponentially decaying sine fit allows determining the spectral width of the pulses applied in panel **a** (colored areas). **c** The spin coherence time is measured by applying a three-pulse spin-echo sequence (inset). The echo amplitude decreases with increasing sequence length and is well-fit with a Gaussian decay from which $T_2 = 48(2)$ $\mu$s is obtained at a temperature of 2.75 K (bright red). At 4.45 K, a reduced value of $T_2 = 23(1)$ $\mu$s is observed (dark red). Error bars: 1 SD.

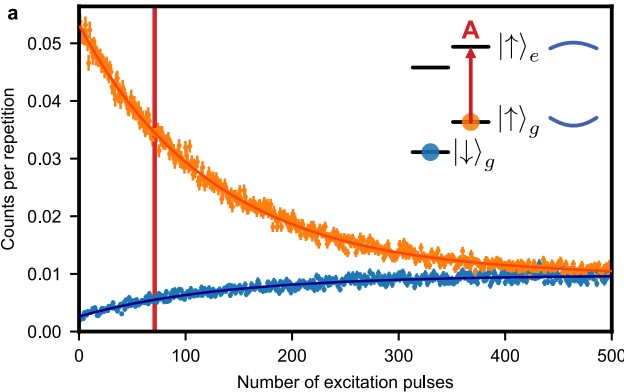

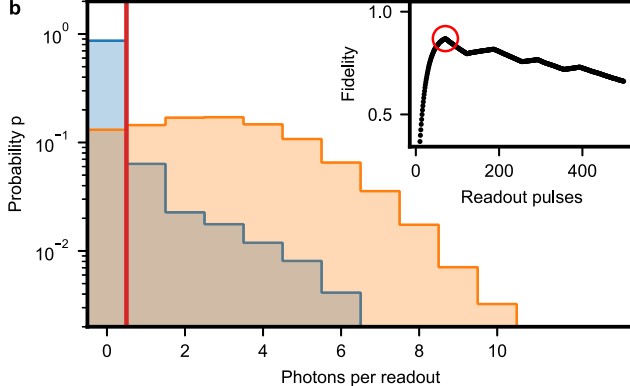

**Fig. 5 | Spin cyclicity and single-shot readout. a** The spin is probed on the spin-preserving transition (red arrow in the level scheme in the inset), which is resonant with the cavity (blue curved lines), after initialization in the bright (orange) or dark (blue) state. An exponential fit to the average fluorescence (solid lines) allows determining the cyclicity ζ. Error bars: 1 SD. **b** After initialization, 71 pulses (red line in panel **a** and red circle in the inset) are applied on the spin-preserving line to read out the spin state. The probability distributions for the number of detected photons are clearly separated for the two initial spin states (orange and blue), thus allowing for readout with a fidelity of 86.9(8)% using a threshold photon number of one (red line). Inset: Single-shot readout fidelity depending on the number of readout pulses while choosing the optimal photon threshold.

cyclicity ζ, which measures how often a photon is emitted on average on the readout transition before the spin is flipped[43].

This is achieved at the chosen magnetic field of $B = 0.3$ T when the cavity is tuned on resonance with the spin-preserving transition A. Then, the spin-flip transition D out of the bright excited state $|\uparrow\rangle_e$ is detuned by 3.6 GHz, i.e. ≈ 1.5 cavity linewidths, such that it experiences a ten-fold lower Purcell enhancement[20] when assuming an identical dipole projection to the cavity mode. The resonator thus enhances ζ significantly as compared to its free-space value, which facilitates spin readout. Albeit larger B-fields may further increase ζ, they would also reduce the spin lifetime owing to the $B^5$ dependence of spin-lattice relaxation via the direct process[27].

To characterize the single-shot readout, we initialize the spin by optical pumping and then apply a readout sequence comprising 500 laser pulses of 0.02 μs duration on the "A" transition. First, we determine the cyclicity. To this end, the number of detected photons in 3 μs intervals after each pulse is averaged over 25000 iterations, see Fig. 5a. When the spin is initialized in the bright state $|\uparrow\rangle_g$ (orange), we obtain a high fluorescence level after the first readout pulses. However, upon repeated excitation, the spin can flip to the dark state. Thus, the signal decays exponentially with a decay constant of $N_\uparrow = 127(2)$ pulses. In contrast, if the spin is initialized in the dark state $|\downarrow\rangle_g$, one initially observes a low fluorescence level, which rises within $N_\downarrow = 135(5)$ pulses. Both values are almost identical when considering the standard deviation of the fit. Thus, after calibrating the probability that the used pulse excites the dopant, $p = 0.78(6)$, we can specify an average cyclicity of $\zeta = pN = 103(7)$ for both states.

The observation that also the dark state decays when repeatedly applying readout pulses indicates that the dominating spin-relaxation mechanism is not the optical decay of the bright state via the spin-flip transition D. Instead, the comparable decay rates, $N_\uparrow \simeq N_\downarrow$, suggest the same spin-decay channel for both the bright and the dark state. However, owing to the long lifetime and comparably short interval of 10 μs between subsequent excitation pulses, we can exclude spin-lattice relaxation (see Fig. 3). To study the effect further, we thus vary the power of the readout pulses and measure the cyclicity. As shown in Fig. 6, ζ is reduced with increasing pulse area. The precise mechanism of this drop will require further investigation. For now, the observed decay entails a trade-off: At high power, the readout fidelity will be reduced by the moderate cyclicity. At low power, instead, the readout duration is increased, and therefore detector dark counts will start to play a role.

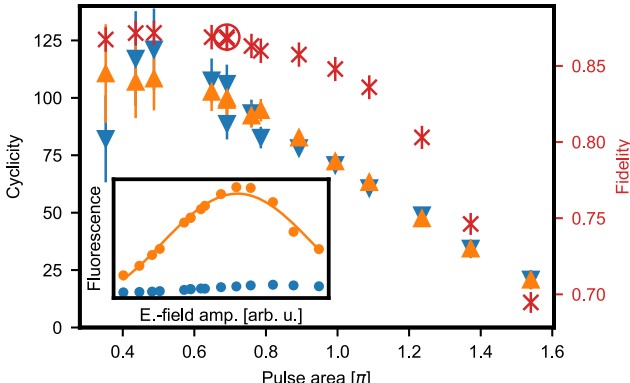

**Fig. 6 | Optimization of the excitation pulses used for single-shot readout.** Inset: The spin is prepared in the bright (orange) or dark (blue) state and the electric field amplitude of the excitation pulses is varied. When averaging the fluorescence in a time interval of 3 μs after the first ten pulses, coherent Rabi oscillations are obtained. Assuming 100% population transfer at the maximum, one can calibrate the excitation probability as a function of the pulse area. Data points are larger than 1 SD. Main panel: The decay of the fluorescence with the number of applied pulses is measured using the same sequence as in Fig. 5a for different excitation pulse areas. The constant $N_0$ extracted from a fit to the observed exponential decay is then multiplied by the calibrated excitation probability for each pulse amplitude to determine the cyclicity ζ. For both initializations, a reduction of ζ with increasing pulse area is observed, which indicates the presence of an excitation pulse induced spin-flip mechanism that is independent of the initial spin state (blue and orange triangles). Thus, also the single-shot readout fidelity (red, right axis) depends on the used pulse area. The highest value is obtained at higher ζ, i.e. lower power. However, also the fluorescence signal is reduced with lower power, which entails a longer readout duration and a stronger contribution of detector dark counts. Error bars: 1 SD.

We find that the intermediate power used in the above cyclicity measurement, Fig. 5a, is a good compromise. Thus, using the same data set we continue with the analysis of the readout fidelity $F$, which is given by the probability to correctly detect the intended spin state. As this can differ for the two spin states, $F_\downarrow \neq F_\uparrow$, we use the smaller value as a lower bound: $F \equiv \min(F_\downarrow, F_\uparrow)$. To study the dependence of $F$ on the pulse number $N$, we individually register all detected photons during each single-shot readout sequence. Thus, we can determine the value of $N$ that gives the best $F$ when adapting the photon threshold accordingly. The result is shown in the inset of Fig. 5b, where the kinks

in the curve occur at positions where the optimal threshold photon number is adapted. Using $N = 71$ readout pulses and a threshold of one detected photon, we achieve $F = 86.9(8)\%$. For these conditions, the resulting histograms for both states are shown in Fig. 5b.

While errors in both the initialization and the readout contribute to a reduction of $F$, based on the histograms — in particular the continuous shape of the bright state distribution (orange) at low photon numbers — we find that readout errors dominate. Similarly, due to the short optical lifetime achieved in our experiment, detector dark counts ($\approx 10$ Hz) only lead to a minor reduction of $\approx 0.3\%$. Instead, the extracted single-shot readout fidelity is limited by the high probability of zero-photon-detection events in the bright state, and spin-flips in the dark state — in other words by $\zeta$. Thus, $F$ is reduced for higher pulse powers as shown in Fig. 6 (red crosses). In contrast, a lower excitation probability can lead to a slightly higher fidelity at the cost of a longer readout duration. For technical reasons, the excitation pulses in the shown measurements are repeated every $10\,\mu s$, such that the readout with 71 pulses takes 0.71 ms. Using the maximum rate allowed by the integration interval, this would be reduced to 0.22 ms.

In future experiments, the fidelity and speed of the readout can be further increased by optimizing the photon outcoupling. In addition, one may attempt aligning the optical dipoles using a vector magnet[43], and further enhancing the quality factor of the cavity, where even a hundredfold increase seems feasible with improved nanofabrication[44]. The latter would enable other readout techniques, e.g. via cavity reflection measurements[20], that can fully eliminate photon scattering[45] and thus overcome the limitation currently imposed by the finite cyclicity. We therefore expect that fidelities on par with other solid-state emitters[40], and potentially even approaching common error correction thresholds[46] can be achieved in future devices.

## Discussion

In summary, we have demonstrated the optical single-shot readout of a spin qubit in silicon, which constitutes a major leap towards scalable, integrated and optically controlled quantum information processing devices. To this end, the next key step will be the generation of spin-photon entanglement, which will be feasible in our setup using established cavity-enhanced protocols[20]. With this, our approach will find applications in two main fields: distributed quantum information processing and quantum networking.

For quantum computing, large photonic cluster states can be generated using a single efficient spin-photon interface[47,48]. These can be used as a resource for measurement-based computation[49] which will heavily benefit from the complete quantum photonics toolbox that is readily available in silicon[50]. For scalable spin-based computation, a further optimization of the readout speed and fidelity will be required, with the goal to push it beyond the fault-tolerance threshold. This calls for resonators with higher quality factor[44], and for increasing the photon detection probability by optimizing the resonator and off-chip coupler designs, or by directly integrating efficient single photon detectors[13] and modulators[51] on the same chip.

For quantum networking, the demands on the readout fidelity are somewhat relaxed, as quantum repeaters of the first generation can tolerate larger error probabilities[52,53]. Instead, other properties of the system are more important. In particular, the spin coherence time should be long enough to keep a qubit while a photon entangled with it is propagating. The observed $48(2)\,\mu s$ already corresponds to a travel distance exceeding 10 km of optical fibers, and even longer spin coherence times are expected under dynamical decoupling[42] and in isotopically purified silicon, where recent experiments have demonstrated ms-long coherence with Er:Si ensembles[41]. With this, our system can make best use of one of its key strengths — its emission in the telecom C-band, where the transmission of $\approx 1\%$ after 100 km of optical fibers exceeds that of all other known coherent emitters by more than an order of magnitude[20], offering unique prospects for long-distance

spin-spin entanglement. This will require spectral stability of the emitted light on the timescale of the entanglement protocol. Ensemble measurements have demonstrated a lower bound of 0.03 ms of optical coherence with Er:Si in site A[34], which would be sufficient to overcome the detrimental effect of spectral diffusion using a recently demonstrated protocol[54]. Combined with its potential for commercial fabrication of suited nanophotonic resonators[55] and for spectral multiplexing, i.e. simultaneously controlling several emitters in the same resonator[32,38], our system thus has a unique potential for the upscaling of quantum networks and distributed quantum information processors.

## Methods

### Sample

The nanofabrication process of the erbium-doped photonic crystal resonators follows similar recipes to those described in refs. 17,34. First, the Czochralski-grown silicon device layer of a commercially available silicon-on-insulator wafer (SOITEC) is homogeneously implanted with erbium dopants (Innovion). According to the facility, the isotopes $^{166}Er$, $^{167}Er$ and $^{168}Er$ are implanted at approximately equal concentration. In order to maximize the fraction of emitters with a high Purcell enhancement, the implantation is performed with an energy of 250 keV and a dose of $1 \times 10^{11}$ cm$^{-2}$, resulting in a Gaussian depth profile centered in the 220 nm silicon device layer, with a simulated straggle of $\approx 20$ nm and a peak Er concentration of $\approx 1 \times 10^{16}$ cm$^{-3}$.

Following implantation, the wafer is diced into $10 \times 10$ mm pieces and then annealed at 800 K with a ramp duration of 1 min from room temperature and a hold time of 0.5 min. Subsequently, nanophotonic resonators are patterned using electron-beam lithography (Nanobeam Ltd., nb5) using a positive-tone resist (ZEP 520A), and transferred into the device layer by reactive ion etching (Oxford PlasmaPro 100 Cobra ICP RIE 100) in a cryogenic fluorine chemistry. Finally, the BOX layer below the nanostructures is removed using a buffered HF etch. In order to compensate for fluctuations in the nanofabrication process, we make many resonators on each chip with a sweep of their design frequency. We then select the one that is closest to the erbium transitions in site A after cooling down to cryogenic temperatures. To precisely tune this cavity on resonance, we first condense a layer of argon ice onto the sample and then evaporate parts of it in a well-controlled way using resonant laser fields with a power of $\approx 0.1$ mW. Furthermore, we control the sample temperature to fine-tune the cavity resonance, which exhibits a blue shift of about 4 GHz when increasing its temperature from 1.8 K to 4.5 K. This effect is attributed to the condensation of a liquid helium thin film at the surfaces of the nanostructures in the exchange gas atmosphere of the used cryostat (Attocube AttoDry 2100).

### Experimental setup

The sample is mounted on a three-axis nanopositioning system (Attocube ANPx312, ANPz 102). Optical pulses are generated from continuous-wave laser systems (Toptica CTL or NKT Photonics BASIK X15). To this end, a single-sideband modulation setup is used, comprising an optical IQ modulator (iXblue MXIQER-LN-30) and a modulator bias controller (iXblue MBC-IQ-LAB-A1), which allows for frequency sweeps with a range of several GHz, as well as two acousto-optic modulators (Gooch&Housego Fiber-Q) for enhancing the on-off contrast. The experimental sequences controlling these devices are implemented on two arbitrary waveform generators (Zurich Instruments HDAWG and SHFSG). The light is then coupled onto the photonic chip by an adiabatic coupler consisting of a tapered waveguide and a tapered optical fiber. The light reflected from the cavities or emitted by the dopants is separated from the input using a beamsplitter with a splitting ratio of 95:5 (Evanescent Optics Inc.). It is detected with a superconducting nanowire single-photon detector (ID

Quantique), with a detection efficiency of 80(5)% and a dark count rate of 10 Hz, located in a second cryostat. To prevent the detector from latching due to the excitation pulses, a fast optical switch for gating is employed (Agiltron Ultra-fast Dual Stage SM NS 1 × 1 Switch, rise time: < 100 ns, transmission: 78%) which is also controlled by the AWGs. Microwave pulses are applied via a copper wire at a distance of ≈ 2 mm to the chip using an amplifier with a maximum output power of 100 W (MiniCircuits ZHL-100W-352+).

## Data availability
The datasets generated during and analyzed in this current study have been deposited in the mediaTUM database under the following https://doi.org/10.14459/2024mp1760064[56].

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

## Acknowledgements

This project received funding from the European Research Council (ERC) under the European Union's Horizon 2020 research and innovation program (grant agreement No 757772), from the Deutsche Forschungsgemeinschaft (DFG, German Research Foundation) under Germany's Excellence Strategy - EXC-2111 - 390814868, from the German Federal Ministry of Education and Research (BMBF) via the grant agreements No 16KISQ046 and 13N16921, and from the Munich Quantum Valley funded by the Bavarian state government via the Hightech Agenda Bayern Plus.

## Author contributions

A.G. and J.P. fabricated the sample. A.G. and A.U. performed the measurements. A.G., A.U., and A.R. analyzed and interpreted the data. A.R. supervised the study. A.G. and A.R. wrote the manuscript.

## Funding

## Competing interests

The authors declare no competing interests.
