## [Transparent Peer Review file · Nature Communications]

Optical single-shot readout of spin qubits in silicon

Corresponding Author: Dr Andreas Reiserer

Version 0:

Reviewer comments:

Reviewer #1

(Remarks to the Author)

The paper "Optical single-shot readout of spin qubits in silicon" reports on a study of individual erbium ions embedded in a silicon nanophotonic cavity, mainly their spin properties and their single-shot readout through optical methods.

The paper presented three main experimental results. The first is a spin-photon interface. A single Er was confirmed by its g_2 measurement, and the spin structure was revealed by a pump-probe spectrum measurement when an external magnetic field was applied. The second is the spin property study. They report a spin T_1 of 0.44s and a T_2 (Hahn) of 48us. A strongly coupled silicon-29 nuclear spin is believed to be identified in the measurement. The third one is the optical single-shot readout. They studied the parameter regime of the single-shot measurement and realized a readout fidelity of 86.9%.

Single-shot readouts of erbium spins have been demonstrated when embedded in different hosts using different methods, including in YSO and CaWO₄ using optical methods, and in silicon using charge sensing techniques. The former benefits from its compatibility with quantum working infrastructure, whereas the latter benefits from using silicon, the most developed semiconductor material, and therefore access to silicon photonics and quantum computing platforms. This paper combines the advantage of using a silicon substrate and efficient optical readout. It presents an important step towards silicon-based quantum networks and distributed quantum information processing.

The paper is very well written. Clear experimental protocol and data analysis are provided. The main conclusions are supported by solid experimental results and analysis. The results are of very high interest to the quantum information processing and networking community. For these reasons, I recommend this paper for publication in Nature Communications, after the authors have addressed the following comments/questions.

1. The Purcell enhanced Er emission has a short lifetime of 0.8us, due to the intrinsic short radiative lifetime given by the silicon host and a decent enhancement given by the nanophotonic cavity. To my best knowledge, this is the fastest Er emission being demonstrated. Could the authors confirm/correct me?
2. What makes the red ion in Fig. 1b the candidate for the following investigation? The rightmost peak in the spectrum seems to have a narrower linewidth, similar brightness, and well-isolated spectrum.
3. In Fig. 2b, when the probe hits transition A or B, the PL signal gets (1) stronger when the pump and probe drive different spin states and (2) weak when they drive the same spin state. However, why is the signal stronger than in case (2) if the probe is detuned from the transitions? I would expect background noise for both cases.
4. In Fig. 2b, transition lines A and B are significantly resolvable and line C is only vaguely resolvable because of its non-spin-preserving nature (probably also due to different purcell enhancement). Why is D unresolvable? In addition, there is another vaguely resolvable line at -0.2GHz. What is the potential origin of that?
5. What is the dashed diagonal line in Fig. 2b?
6. In lines 123 to 125, the origin of the ODMR splitting is attributed to superhyperfine splitting to a nearby Si-29 nucleus. However, some other possibilities, such as Er-Er coupling, cannot be immediately excluded. Although definite proof and further investigation might be beyond the scope of this paper, I believe a discussion of the potential reasons and qualitative estimation of them will consolidate the hypothesis.
7. In Fig. 3a, it would be helpful to include a proposed superhyperfine energy diagram.
8. An analysis of the different Rabi frequencies for different spin resonances in Fig. 3b (even qualitatively) will be inspirational.
9. In Fig. 4b's inset, the readout fidelity oscillates as a function of the pulse number. What causes this oscillation?
10. Could the authors estimate the total amount of erbium ions using the knowledge of implantation density and their cavity's mode profile? My rough estimation is $1\mu\text{m}^2 * 10^{11} \text{ cm}^{-2}$, resulting in around 1000 ions. Given that around 10 ions are observed within site A's inhomogeneous linewidth, could one conclude that the yield of site A using the implantation and

annealing recipe in this paper is at a 1% level?

11. As described in Method 1, the fine-tuning was done by tuning the cryostat's temperature. What is the temperature for different measurements? The spin T2 was measured at 2.75K and 4.45K, which corresponds to a significant cavity shift of 2.5GHz. What measures were taken to make the two datasets comparable?

12. As one of the main selling points of the paper is the silicon platform, I suggest the authors include a summary of competing systems to show their pros and cons.

Reviewer #2

(Remarks to the Author)

In this manuscript, the authors report the coherent control, T2 measurement and optical single-shot readout (SSRO) of the electron spin of a single erbium dopant in silicon.

Using Hahn echo sequence, they measure a coherence time of roughly 50 μ s, that could be easily increased up to 1 ms by using isotopically purified ^{28}Si (Ref. 45). To enable the electron spin SSRO, the emitter-cavity coupling is tuned to have the emission of only one of the 2 spin-conserving transitions to be strongly enhanced by Purcell effect. For a first demonstration, the achieved readout fidelity is already quite good (~87%).

So far, the only published paper addressing the spin control of single emitters in silicon is Higginbottom et al., Nature 2022 (Ref. 29), presenting electron spin initialization of a single T center by optical pumping.

Compared to this previous article, the current manuscript has reached several milestones in developing a spin-photon interface in silicon, namely Rabi oscillations, Hahn echo and optical SSRO on a single electron spin. This last demonstration is particularly impressive as it requires at the same time high photon collection and strongly cycling optical transitions under resonant excitation. Together with their direct emission in the telecom C-band, these results place Er dopants as very strong candidates for quantum networks based on integrated silicon devices.

As recalled in the manuscript, the literature on the control of erbium dopants is vast, and includes the optical single shot spin readout on a single ion (Ref. 12). However, these experiments were carried out in other materials, mostly YSO, which cannot rival silicon in terms of nanofabrication maturity and scalability prospects. Given the maturity of silicon photonic chips, similar progress to that seen with erbium dopants in YSO could be expected in silicon in the near-future.

The manuscript is well written and very pleasant to read. Data are clear, analyzed and discussed, with experimental details provided.

I only have 3 minor comments and questions:

- I didn't get why the authors have not shown the $g(2)$ plot measured on the most red-detuned dopant, especially with such low antibunching ($g(2)(0) < 2\%$). It would strengthen the manuscript to show it, even in Extended data figures.
- The delay between optical pulses is given at the end of page 7 (line 206). This information could be useful earlier when discussing the spin-relaxation mechanisms limiting the cyclicity.
- Could the authors indicate which Er isotope has been implanted?

In conclusion, for sure this manuscript deserves publication in Nature Communications journal. In my opinion, the results presented here could even target higher impact journals.

Version 1:

Reviewer comments:

Reviewer #1

(Remarks to the Author)

The authors' responses have addressed my questions and concerns very well. Therefore, I recommend that this paper be published in Nature Communications.

Reply to the referee comments on NCOMMS-24-38872-T

We thank both referees for their positive evaluation of our work. In the following, we will give a detailed reply to their comments (marked in blue) and all changes to the manuscript (marked in red).

Report of the First Referee -- NCOMMS-24-38872-T:

The paper “Optical single-shot readout of spin qubits in silicon” reports on a study of individual erbium ions embedded in a silicon nanophotonic cavity, mainly their spin properties and their single-shot readout through optical methods.

The paper presented three main experimental results. The first is a spin-photon interface. A single Er was confirmed by its g_2 measurement, and the spin structure was revealed by a pump-probe spectrum measurement when an external magnetic field was applied. The second is the spin property study. They report a spin T_1 of 0.44s and a T_2 (Hahn) of 48us. A strongly coupled silicon-29 nuclear spin is believed to be identified in the measurement. The third one is the optical single-shot readout. They studied the parameter regime of the single-shot measurement and realized a readout fidelity of 86.9%.

Single-shot readouts of erbium spins have been demonstrated when embedded in different hosts using different methods, including in YSO and CaWO₄ using optical methods, and in silicon using charge sensing techniques. The former benefits from its compatibility with quantum working infrastructure, whereas the latter benefits from using silicon, the most developed semiconductor material, and therefore access to silicon photonics and quantum computing platforms. This paper combines the advantage of using a silicon substrate and efficient optical readout. It presents an important step towards silicon-based quantum networks and distributed quantum information processing.

The paper is very well written. Clear experimental protocol and data analysis are provided. The main conclusions are supported by solid experimental results and analysis. The results are of very high interest to the quantum information processing and networking community. For these reasons, I recommend this paper for publication in Nature Communications, after the authors have addressed the following comments/questions.

We thank the First Referee for the accurate summary of the main experimental results presented in this manuscript and the expected impact for the field, and highly appreciate the recommendation for publication in Nature Communications.

1. The Purcell enhanced Er emission has a short lifetime of 0.8us, due to the intrinsic short radiative lifetime given by the silicon host and a decent enhancement given by the nanophotonic cavity. To my best knowledge, this is the fastest Er emission being demonstrated. Could the authors confirm/correct me?

We thank the reviewer for this remark and can confirm that this is the fastest optical emission reported for a single erbium dopant. To further emphasize this is the text, we modified the corresponding paragraph which now reads:

“The obtained optical lifetime of 0.803(11) μ s represents the fastest optical emission demonstrated for a single erbium dopant, almost ten-fold faster than that observed in previous experiments using other host crystals [16, 17]. This will speed up all operations of the spin-photon interface, including optical spin readout and spin initialization.”

2. What makes the red ion in Fig. 1b the candidate for the following investigation? The rightmost peak in the spectrum seems to have a narrower linewidth, similar brightness, and well-isolated spectrum.

In the submitted manuscript, we mentioned that: *“All measurements are then performed on the most red detuned dopant (red) that exhibits the highest single photon purity in this device $g(2)(0) = 0.019(1)$ ”*. To further clarify the choice of emitter, we now mention: *“The initial characterization of the device follows the techniques introduced in [10] and described in the Methods (A.3). The resulting spectrum is shown in Fig. 1b. For a more detailed investigation of the individual emitters, the cavity is tuned precisely on resonance with each of them by varying the temperature (Methods A.1). By comparing their properties (Methods A.4), we find that the most red-detuned dopant (red) simultaneously exhibits the highest single-photon purity, $g(2)(0) = 0.019(1)$, and the largest Purcell enhancement, $P = 177(2)$ in comparison to the bulk [41], which in turn results in the most favorable ratio between the lifetime-limited linewidth and the spectral diffusion linewidth among the dopants in this device. All measurements in this work are thus performed on this emitter.”*

We furthermore added an additional short paragraph to the methods section:

“A.4 Choice of the dopant

As shown in Fig.1b, three erbium emitters in the studied device are spectrally isolated from the others and can thus be individually addressed [15, 17]: the most red-detuned dopant (in the following: red), the most blue-detuned dopant (blue) and the dopant at a detuning of about 0.12 GHz (center). We find that these three isolated emitters have Lorentzian spectral-diffusion linewidths of 47(1) MHz (red), 33(2) MHz (center) and 13.5(5) MHz (blue) FWHM. The cavity is tuned on resonance with each of these emitters using the temperature tuning scheme described in part A.1 for additional characterization measurements. We find that the red dopant exhibits the best single-photon purity, $g(2)(0) = 0.019(1)$, while the others have slightly worse values of 0.07(1) and 0.09(1) for the center and blue emitter, respectively. Furthermore, the red dopant shows the strongest Purcell enhancement, $P = 177(2)$, compared to values of 89(2) (center) and 29(1) (blue). The observed variation of the Purcell enhancement between the different emitters is expected because of their integration at random positions within the cavity mode [10]. As a result, the ratio between the lifetime-limited linewidth and the spectral diffusion linewidth is superior for the red dopant. Combining all aspects, the red dopant exhibits the most promising properties in this device and is thus used throughout this work.”

3. *In Fig. 2b, when the probe hits transition A or B, the PL signal gets (1) stronger when the pump and probe drive different spin states and (2) weak when they drive the same spin state. However, why is the signal stronger than in case (2)*

if the probe is detuned from the transitions? I would expect background noise for both cases.

From the comment, we understand that the measurement needs further clarification. To this end, we modified the caption: “**Relative fluorescence when the emitter is irradiated with pump and probe pulses of different frequency. Compared to the average signal,** the fluorescence after resonant probe pulses is enhanced (red) or reduced (blue) by preceding pump pulses if the frequency of the latter is on resonance with one of the four optical transitions.”

In addition, we added a quantitative description of the signal to the main text: “To determine the frequencies of the relevant transitions, we employ a pump-probe scheme with 500 consecutive pump pulses of 0.1 μs duration followed by a single 0.2 μs long probe pulse, **after which we record the fluorescence signal S [...]. For every probe laser frequency, we then average over all pump frequencies to determine the average signal S_a [...]. In Fig. 2b we then show the pump-induced fluorescence difference $S_d = S - S_a$ [...].”**

With the detailed explanation, it now becomes clear that one is looking at the difference induced by the pump, which leads to a depletion and thus a negative signal in case (2).

4. *In Fig. 2b, transition lines A and B are significantly resolvable and line C is only vaguely resolvable because of its non-spin-preserving nature (probably also due to different Purcell enhancement). Why is D unresolvable? In addition, there is another vaguely resolvable line at -0.2GHz. What is the potential origin of that?*

We added: “**On the spin-flip transitions C and D, no significant signal is obtained after the probe pulses, in agreement with the expectation from the spin Hamiltonian [Holzäpfel 2024]. This explains the absence of peaks in the right side of the figure. The faint features on the left side (around -0.2 and -0.4 GHz, close to transition C) are instead attributed to other emitters because they do not exhibit the characteristic pattern for all four pump frequencies.**”

5. *What is the dashed diagonal line in Fig. 2b?* We added the following explanation to the caption of the figure: “**As a guide-to-the-eye, the dashed black diagonal line indicates identical pump and probe frequencies.**”
6. *In lines 123 to 125, the origin of the ODMR splitting is attributed to superhyperfine splitting to a nearby Si-29 nucleus. However, some other possibilities, such as Er-Er coupling, cannot be immediately excluded. Although definite proof and further investigation might be beyond the scope of this paper, I believe a discussion of the potential reasons and qualitative estimation of them will consolidate the hypothesis.*

We agree and have thus adapted the text as follows: “The observed line splitting is **attributed to the coupling of the emitter to surrounding spins. At the used low dopant concentration, the estimated coupling to other erbium spins is too small to explain the observations. However, one expects that in our sample with its natural ^{29}Si isotopic abundance, individual nuclear spins exhibit a superhyperfine coupling that is larger than the broadening from the rest of the spin bath. A possible configuration [...]**”

7. *In Fig. 3a, it would be helpful to include a proposed superhyperfine energy diagram.*

We have added the proposed level scheme as Extended Data Figure 2, and introduce it in the main text: “A possible configuration is shown in Extended Data Fig. 3, albeit other configurations cannot be ruled out with certainty based on our measurements.”

8. *An analysis of the different Rabi frequencies for different spin resonances in Fig. 3b (even qualitatively) will be inspirational.*

Unfortunately, with the current data we can only speculate about the origin of this deviation. We thus added: “we obtain Rabi oscillations with pi-pulse lengths of 2.3(2) μs and 1.6(2) μs when the MW is applied on the central peak or on the left side peak, respectively. Further measurements will be required to explain the increased oscillation frequency in the latter case.”

9. *In Fig. 4b's inset, the readout fidelity oscillates as a function of the pulse number. What causes this oscillation?*

To explain the oscillation in the curve, we added: “The result is shown in the inset of Fig. 4b, where the kinks in the curve occur at positions where the optimal threshold photon number is adapted.”

10. *Could the authors estimate the total amount of erbium ions using the knowledge of implantation density and their cavity's mode profile? My rough estimation is $1\mu\text{m}^2 * 10^{11}\text{cm}^{-2}$, resulting in around 1000 ions. Given that around 10 ions are observed within site A's inhomogeneous linewidth, could one conclude that the yield of site A using the implantation and annealing recipe in this paper is at a 1% level?*

We added: “From the observed number of emitters, the resonator mode volume and the implanted dose, we find that less than 1% of the emitters are integrated in site A, consistent with our earlier work [41].”

11. *As described in Method 1, the fine-tuning was done by tuning the cryostat's temperature. What is the temperature for different measurements? The spin T2 was measured at 2.75K and 4.45K, which corresponds to a significant cavity shift of 2.5GHz. What measures were taken to make the two datasets comparable?*

In all measurements, the temperature is chosen such that the cavity is on resonance with the emitter. To clarify that this is also the case for the 4.45 K measurement, we added: “To investigate the source of decoherence, we perform a second measurement at 4.45 K. Owing to condensed gas on the sample that shifts the cavity frequency, we can keep the optical transitions on resonance at this increased temperature. We find that the coherence time differs between the measurements at 2.75 and 4.45 K [...]”

12. *As one of the main selling points of the paper is the silicon platform, I suggest the authors include a summary of competing systems to show their pros and cons.*

We would prefer not to show such a table, as we feel that it would unnecessarily put the pioneering work of many other research groups in a bad light. Instead,

we believe that even without such a table the key advantages of the silicon platform, in participated regarding the upscaling of quantum systems, are made clear in the text.

Report of the Second Referee -- NCOMMS-24-38872-T:

In this manuscript, the authors report the coherent control, T2 measurement and optical single-shot readout (SSRO) of the electron spin of a single erbium dopant in silicon.

Using Hahn echo sequence, they measure a coherence time of roughly 50 μs , that could be easily increased up to 1ms by using isotopically purified ^{28}Si (Ref. 45). To enable the electron spin SSRO, the emitter-cavity coupling is tuned to have the emission of only one of the 2 spin-conserving transitions to be strongly enhanced by Purcell effect. For a first demonstration, the achieved readout fidelity is already quite good (~87%).

So far, the only published paper addressing the spin control of single emitters in silicon is Higginbottom et al., Nature 2022 (Ref. 29), presenting electron spin initialization of a single T center by optical pumping. Compared to this previous article, the current manuscript has reached several milestones in developing a spin-photon interface in silicon, namely Rabi oscillations, Hahn echo and optical SSRO on a single electron spin. This last demonstration is particularly impressive as it requires at the same time high photon collection and strongly cycling optical transitions under resonant excitation. Together with their direct emission in the telecom C-band, these results place Er dopants as very strong candidates for quantum networks based on integrated silicon devices.

As recalled in the manuscript, the literature on the control of erbium dopants is vast, and includes the optical single shot spin readout on a single ion (Ref. 12). However, these experiments were carried out in other materials, mostly YSO, which cannot rival silicon in terms of nanofabrication maturity and scalability prospects. Given the maturity of silicon photonic chips, similar progress to that seen with erbium dopants in YSO could be expected in silicon in the near-future.

The manuscript is well written and very pleasant to read. Data are clear, analyzed and discussed, with experimental details provided.

We thank the Second Referee for the short summary of the main experimental results, his recommendation and the extended comment on the anticipated impact of our work.

I only have 3 minor comments and questions:

1. *I didn't get why the authors have not shown the $g(2)$ plot measured on the most red-detuned dopant, especially with such low antibunching ($g(2)(0) < 2\%$). It would strengthen the manuscript to show it, even in Extended data figures.*

We did not show such a figure as the anti-bunching at zero delay is comparable to that observed in our previous work [Gritsch et al. 2023] that, however, used a different device. However, we agree that it is an important property of the device. Following the comment we thus **included an additional Extended Data Figure that shows the autocorrelation function.**

2. *The delay between optical pulses is given at the end of page 7 (line 206). This information could be useful earlier when discussing the spin-relaxation mechanisms limiting the cyclicity.*

We have adapted the corresponding sentence accordingly: “However, owing to the long lifetime and comparably short interval of 10 μ s between subsequent excitation pulses, we can exclude spin-lattice relaxation (see Extended Data Fig. 1)”

3. *Could the authors indicate which Er isotope has been implanted?*

We did not implant a specific isotope. To clarify, we added: “According to the facility, the isotopes ^{166}Er , ^{167}Er and ^{168}Er are implanted at approximately equal concentration.”